# The Role of Insulin-like Growth Factor Binding Protein (IGFBP)-2 in DNA Repair and Chemoresistance in Breast Cancer Cells

**DOI:** 10.3390/cancers16112113

**Published:** 2024-05-31

**Authors:** Alaa Mohammedali, Kalina Biernacka, Rachel M. Barker, Jeff M. P. Holly, Claire M. Perks

**Affiliations:** 1Cancer Endocrinology Group, Learning and Research Building, Southmead Hospital, Translational Health Sciences, Bristol Medical School, Bristol BS10 5NB, UK; ward8d2@hotmail.com (A.M.); kalina.biernacka02@gmail.com (K.B.); mdrmh@bristol.ac.uk (R.M.B.); 2Translational Health Sciences, Bristol Medical School, Bristol BS10 5NB, UK; jeff.holly@bristol.ac.uk

**Keywords:** IGFBP-2, DNA damage and repair, chemotherapy, DNA-PKcs, breast cancer

## Abstract

**Simple Summary:**

Globally, breast cancer is the most common malignancy and the most frequent cause of cancer-related deaths among women. Chemotherapy is the major systemic treatment for breast cancer, but unfortunately patients often develop resistance, which leads to a poor outcome. Chemotherapy works in a defined manner and this study explores the role of a molecule called insulin-like growth factor binding protein-2 (IGFBP-2), frequently increased by tumours, in the response of cancer cells to chemotherapy. This work suggests that reducing IGFBP-2 has the potential to improve sensitivity to chemotherapy and may provide insight into optimising current treatment strategies.

**Abstract:**

The role if insulin-like growth factor binding protein-2 (IGFBP-2) in mediating chemoresistance in breast cancer cells has been demonstrated, but the mechanism of action is unclear. This study aimed to further investigate the role of IGFBP-2 in the DNA damage response induced by etoposide in MCF-7, T47D (ER+ve), and MDA-MB-231 (ER-ve) breast cancer cell lines. In the presence or absence of etoposide, IGFBP-2 was silenced using siRNA in the ER-positive cell lines, or exogenous IGFBP-2 was added to the ER-negative MDA-MB-231 cells. Cell number and death were assessed using trypan blue dye exclusion assay, changes in abundance of proteins were monitored using Western blotting of whole cell lysates, and localization and abundance were determined using immunofluorescence and cell fractionation. Results from ER-positive cell lines demonstrated that upon exposure to etoposide, loss of IGFBP-2 enhanced cell death, and this was associated with a reduction in P-DNA-PKcs and an increase in γH2AX. Conversely, with ER-negative cells, the addition of IGFBP-2 in the presence of etoposide resulted in cell survival, an increase in P-DNA-PKcs, and a reduction in γH2AX. In summary, IGFBP-2 is a survival factor for breast cancer cells that is associated with enhancement of the DNA repair mechanism.

## 1. Introduction

Breast cancer is the most common malignancy in women, with aggressive forms leading to life-threatening complications despite high survival rates [1]. Deficiencies in DNA damage and repair mechanisms are known risk factors for the development and progression of various cancers, including breast cancer. Double-strand breaks (DSBs) can occur in cells as a result of drug exposure, such as chemotherapy [2]. The drug etoposide, for example, inhibits DNA topoisomerase II, causing DNA strand breaks by blocking DNA re-ligation [3]. Nonhomologous end joining (NHEJ) and homologous recombination (HR) represent the primary mechanisms for repairing double-strand breaks (DSBs), with NHEJ characterised by its efficiency and speed [4]. The activation of NHEJ repair is notably influenced by the phosphorylation activity of DNA-dependent protein kinase (DNA-PK). DNA-PK is a complex comprising DNA-PKcs and the Ku70/Ku80 heterodimer, which binds to DNA ends following damage. DNA-PKcs and ataxia-telangiectasia-mutated (ATM) participate in phosphorylating the histone variant H2AX at Ser139, resulting in the formation of γ-H2AX foci—an indicator of DNA damage and the DNA damage response (DDR) [5]. Given the crucial role of DNA-PKcs in the DDR pathway, cancer therapies specifically target it, along with other agents detrimental to DNA integrity [6]. In prostate cancer, DNA-PKcs plays a specific modulatory role in transcriptional networks that promote cell invasion, migration, and metastasis in both in vitro and in vivo settings, indicating its pivotal role in driving lethal cancer progression [7]. Analysis of clinical samples from human breast cancer revealed that elevated levels of DNA-PKcs were associated with higher tumour grade, a basal-like subtype, and poor prognosis [8]. The IGF-I system plays a crucial role in various aspects of breast cancer, including its development, progression, metastasis, and resistance to cancer treatments, as evidenced by numerous studies [9,10,11,12]. The insulin-like growth factor (IGF) axis comprises two peptides (IGF-I and IGF-II), IGF receptors (IGF-IR and IGF-IIR), and insulin-like growth factor binding proteins (IGFBPs 1–6) [13]. As the effects of IGFBP-2 can occur through either IGF-1 receptor-dependent or -independent actions [14,15], understanding how IGFBP-2 influences the DNA damage response holds significance for clinical trials using IGF-IR inhibitors alongside DNA-damaging cytotoxic drugs [16]. Numerous studies focusing on breast cancer cells have demonstrated that increased IGFBP-2 levels can intrinsically stimulate cancer cell proliferation and confer resistance to treatment [17,18]. In estrogen-receptor-positive breast cancer cells, such IGF-1R-independent actions of IGFBP-2 were found to be dependent on the presence of ERα, as silencing ERα eliminated these intrinsic effects of IGFBP-2 [19]. In addition, previous research has indicated that IGFBP-2 influences DNA-PKcs in DSB repair (DSBR) in prostate and esophageal cancer cells [20,21,22]. This study aimed to explore whether IGFBP-2 plays a role in the mechanism of DNA damage and repair in breast cancer cells and whether this is dependent or independent of interaction with IGFs.

## 2. Materials and Methods

Recombinant IGFBP-2 (ab63223, 20 µg) was bought from Abcam, Cambridge UK, etoposide (10 g) from Tocris Bioscience, Oxford, UK, and all other chemicals unless otherwise stated were from Sigma, Gillingham, Dorset, UK.

### 2.1. Cell Culture

MCF-7, T47D, and MDA-MB-231 breast cancer cells were purchased from ATCC (American Type Culture Collection, Middlesex, UK) and maintained as described previously [22]. The American Type Culture Collection authenticates using short tandem repeat DNA and the cell lines are confirmed as mycoplasma-negative in our routine quality control.

### 2.2. Treatment with Etoposide

MCF-7 and MDA-MB-231 cells were treated with (40 µM) and T47D (60 µM) etoposide for 24 hrs. We performed rigorous testing to determine the optimum dose for etoposide in all the cell lines. This included an assessment of different doses (40, 60, and 80 µM) and time courses (1, 4, 24, and 48 h) with the measurement of γH2AX abundance as a marker of DNA damage. 

### 2.3. IGFBP-2 Silencing Using siRNA

IGFBP-2 was silenced in MCF-7 and T47D cells using siRNA (Qiagen; target sequence CAGTTCTGACACACGTATTTA) and compared with AllStars non-silencing (NS) siRNA. Effective IGFBP-2 silencing was validated using two different siRNAs as described previously [19].

### 2.4. Cell Counting Experiments Using Trypan Blue Dye Exclusion

Cell growth and death were assessed using Trypan blue dye exclusion cell counting using a hemocytometer. We have previously demonstrated that Trypan blue dye exclusion assay compares with other more specific measures of apoptosis and cell survival including flow cytometry, MTT assay, PARP cleavage, colony formation assay, and morphological assessment [19,23,24,25].

### 2.5. Western Blotting

Western blotting was performed as described before [19]. In brief, 25–30 μg of protein were loaded and run on 4–20% SDS-PAGE gels, transferred to nitrocellulose membrane (BioRad, Watford, Hertfordshire), and probed with the following antibodies: IGFBP-2 (1:1000, ab109284; Abcam, Cambridge, UK), λH2AX, P-DNA-PKcs, and α Tubulin (1:5000 Merck Millipore, Burlington, MA, USA). Depending on the species of the primary antibody, the blots were incubated for 1 h in horseradish peroxidase-linked mouse or rabbit as secondary antibody (1:2000 Merck Millipore Hertfordshire, UK). The proteins were assessed using Clarity ECL substrate (BioRad, Hertfordshire, UK) using BioRad Chemidoc XRS + system and the images were analysed using Image J Lab software, following the manufacturer’s instructions (BioRad, 170-8265).

### 2.6. Immunofluorescence Staining and Confocal Imaging

Immunofluorescence and confocal imaging were conducted as previously described [26]. In brief, cells were cultured on coverslips in 12-well plates using DMEM containing 5% fetal bovine serum and incubated for 24 h, fixed for 20 min using 4% paraformaldehyde, followed by treatment for 30 min with a 0.5% solution of Triton X to permeabilise the cells. Non-specific binding was blocked with 1 ml of 5% normal goat serum (NGS; Vector Laboratories, Burlingame, CA, USA) for 1 h at room temperature. Then cells were treated with primary antibodies (1:200) overnight at 4 °C, followed by secondary antibodies (1:400) for 1 h at room temperature. Next, 500 µL of 5% NGS were added to the coverslips followed by incubation of the fluorochrome-conjugated secondary antibodies, a goat anti-mouse secondary antibody (Alexa Fluor 594), or/and a goat anti-rabbit secondary antibody (Alexa Fluor 488); 1:400, respectively (Thermo Fisher Scientific, Loughborough, UK). Slides were mounted and counterstained using 4’,6-diamidino-2-phenylindole dihydrochloride (DAPI) (Vector Laboratories, Burlingame, CA, USA). To prevent exposure to direct light, the mounted slides were covered and stored at 4–8 °C. Leica DMi8 inverted epifluorescence microscope and Leica SP8 AOBS confocal laser scanning microscope were used to visualise the slides at the Wolfson Bioimaging facility, University of Bristol. Immunofluorescence images were analysed using a Fiji-ImageJ program that measures the fluorescence intensity of each protein inside the nucleus (ImageJ win-64).

### 2.7. Statistical Analysis

The data were analysed using GraphPad Prism 9.0.1 software and presented as the mean ± SEM of three independent experiments. IBM SPSS Statistics was used for statistical analysis (IBM Corporation, version 28.0.0.0). A *t*-test for independent samples was performed to compare the means of the two groups. The least significant difference (LSD) post hoc test was used after one-way analysis of variance (ANOVA) to compare three or more groups. A statistically significant difference was present at *p* < 0.05 (where * *p* < 0.05, ** *p* < 0.01 and *** *p* < 0.001).

## 3. Results

### 3.1. The Role of IGFBP-2 in DNA Damage in ER Positive Cells

ER-positive MCF-7 and T47D cells secrete high levels of IGFBP-2 and so, to manipulate the levels to determine the role of IGFBP-2 in DNA repair, siRNA silencing of IGFBP-2 was performed. Silencing IGFBP-2 alone increased death by 8.8% (*p* < 0.001) and etoposide alone increased death by 6.4% (*p* < 0.001) compared with the control in MCF-7 cells. In combination, an additive increase in death of 15.8% (*p* < 0.001) (Figure 1A) was observed. Exposure to etoposide was associated with an increase in P-DNA-PKcs levels, which was decreased when IGFBP-2 was silenced (*p* < 0.001). Silencing IGFBP-2 or the presence of etoposide each alone increased levels of γH2AX abundance, and silencing IGFBP-2 with etoposide had an additive effect (Figure 1B,C). Similar data, in addition to immunofluorescence, were obtained with an additional ER-positive breast cancer cell line, the T47D cells (Figure 1D–F and Figure 2). The immunofluorescence data confirm that nuclear abundance of P-DNA-PKcs and γH2AX is increased by etoposide. When IGFBP-2 is silenced, etoposide-induced P-DNA-PKcs is reduced and γH2AX is enhanced (Figure 2).

### 3.2. The Effect of IGF-I on the Role of IGFBP-2 in DNA Damage in MCF-7 Cells

Overall levels of cell death induced following IGFBP-2 siRNA, etoposide, and the combination were reduced in the presence of exogenous IGF-I. However, a similar pattern existed such that silencing IGFBP-2 in combination with etoposide still caused an additive increase in cell death with or without IGF-I (Figure 3A). This pattern was also reflected in the levels of γH2AX and P-DNA-PKcs, whereby the additive effect of IGFBP-2 siRNA and etoposide (in comparison to either alone) caused enhanced γH2AX and reduced P-DNA-PKcs, that was also observed in the presence of IGF-I albeit at a lower level (Figure 3B,C). Further, cell growth was significantly inhibited in the presence of IGFBP-2 siRNA (*p* < 0.01) alone but not by etoposide alone and there was no additive effect in combination. This pattern was unaffected in the presence of IGF-I (Appendix A). Using cellular fractionation, IGFBP-2 was observed basally in both the cytoplasm and nuclear extracts (Figure 3D), and this was also noted using immunofluorescence. Immunofluorescence and OD analysis confirmed that IGFBP-2 was effectively reduced in the presence of IGFBP-2 siRNA (Figure 3E,H). Immunofluorescence and OD analysis also showed that the additive impact of etoposide and IGFBP-2 siRNA on increasing γH2AX and reducing PDNA-PKcs was similarly observed in the presence or absence of IGF-I (Figure 3F–H).

### 3.3. The Role of IGFBP-2 in DNA Damage in ER-Negative MDA-MB-231 Cells

The MDA-MB-231, ER-negative cells, in contrast to the MCF-7 and T47D ER-positive cells, have relatively lower levels of endogenous IGFBP-2 and so exogenous IGFBP-2 was added as opposed to silencing IGFBP-2. Adding exogenous IGFBP-2 resulted in a decrease in cell death induced by etoposide, indicating its role as a survival factor. As shown in Figure 4A, IGFBP-2 reduced the amount of etoposide-induced cell death at both concentrations, with a significant (*p* < 0.001) decrease shown at 250 ng/mL. Figure 4B,C demonstrates that treatment with 250 ng/mL of IGFBP-2 alone had no discernible impact on γH2AX levels. However, in the presence of the chemotherapy, it attenuated etoposide-induced γH2AX levels and significantly elevated levels of P-DNA-PKcs (*p* < 0.001). The levels of IGFBP-2 in the cell supernatants showed a similar pattern to that observed in the lysates, clearly indicating the dose-dependent addition of the exogenous IGFBP-2 (Appendix A).

### 3.4. The Effect of an RGD-Containing Peptide on the Role of IGFBP-2 in DNA Damage in MDA-MB-231 Cells

To investigate whether the actions of IGFBP-2 were integrin-mediated, an RGD-containing peptide was used. Both exogenous IGFBP-2 and RGD significantly reduced etoposide-induced cell death, but it was not reduced further when IGFBP-2 and RGD were added in combination (Figure 5A). There were no significant reductions in total cell number with any treatment (Appendix A). In the absence of etoposide, RGD and exogenous IGFBP-2 had no effect on γH2AX, while etoposide alone increased γH2AX levels and increased those of P-DNA-PKcs significantly. RGD and IGFBP-2 each in combination with etoposide resulted in a reduction in γH2AX and an increase in abundance of P-DNA-PKcs (Figure 5B,C(i,ii)). There was no additive effect when RGD and IGFBP-2 were added in combination with etoposide (Figure 5B,C(iii)). Immunofluorescence indicated that IGFBP-2 was effectively added and that it was present in both the nuclear and cytoplasmic areas of the cells. The pattern of changes in levels of γH2AX and P-DNA-PKcs following treatment was similarly observed using IF (Figure 6A,B).

## 4. Discussion

Despite improvements in breast cancer diagnosis and treatment, resistance to therapy and tumour recurrence occur frequently, with metastasis accounting for 20–30% of breast-cancer-related deaths [27,28]. IGFBP-2 has been reported to modulate oncogenic processes such as proliferation, survival, invasion, metastasis, and angiogenesis [29], and can achieve this by acting in an IGF-dependent or -independent manner. IGF-dependent actions occur by its binding to IGFs and modulating their activity, either by promoting or inhibiting their actions. IGFBP-2 can also act intrinsically by interacting with integrin receptors, which can involve regulation of downstream effectors such as the tumour suppressor genes PTEN, STAT3, and NFκB [30,31,32]. Additionally, IGFBP-2 has the capacity to enter the nucleus, interact with nuclear protein complexes, and promote the production of oncogenic proteins, such as VEGF [33]. Both IGF-dependent and intrinsic actions of IGFBP-2 are often context-dependent.

We previously showed that IGFBP-2 contributes to chemoresistance in both prostate and breast cancer cells [34,35], but the mechanism was unclear. With ER-positive breast cancer cells, silencing IGFBP-2 resulted in enhanced etoposide-induced death, confirming its role as a survival factor. We noted that the loss of IGFBP-2 alone induced DNA damage, indicated by an increase in levels of γH2AX that was exacerbated in the presence of etoposide. Notably, we also observed that silencing IGFBP-2 was associated with reduced levels of P-DNA-PKcs, which is in turn associated with a reduced capacity to repair DNA, thereby enhancing chemosensitivity. This finding aligns with existing research indicating that cells deficient in DNA-PKcs exhibit sensitivity to DNA-damaging therapies [36] and, conversely, that the overexpression of DNA-PKcs in malignancies can confer resistance to DNA damage [37].

We reported that exogenously added IGF-I was still able to act as a survival factor, in the presence or absence of IGFBP-2, but the pattern of cell death was the same: enhanced cell death with the combination of etoposide and IGFBP-2 silencing. This is reminiscent of data we published previously on MCF-7 and T47D cells, in which exogenously added IGF-II, unlike estradiol, could promote cell proliferation in the presence or absence of IGFBP-2. The lack of response to estradiol was due to the loss of ER-α that was evident when IGFBP-2 was silenced. Conversely, the addition of exogenous IGFBP-2 caused an increase in ER-α [19,23,30]. Intriguingly, interesting associations have previously been reported between DNA-PKcs and the ER [38,39,40]. According to these investigations, DNA-PKcs phosphorylates the ER at Ser118, which increases and stabilises ER-dependent transcriptional activity. Additionally, the research indicated that estrogen induces the expression of DNA-PKcs in breast cancer cells, potentially facilitating DNA damage repair [41]. The study also showed that estrogen enhances the interaction between Ku70 and ERα, suggesting that estrogen could boost DNA-PK activity, thereby contributing to its role in DNA repair [38,41]. These reports together with our work may indicate that silencing IGFBP-2, leading to destabilisation of the ER and its subsequent downregulation, could impair the cell’s capacity to repair DNA, via a reduction in DNA-PKcs, in response to etoposide. Another consideration is that there is known crosstalk between IGF and estradiol signalling [42], and so loss of IGFBP-2, by reducing levels of the ER, may also be a mechanism that influences the efficacy of IGF-I in ER-positive cells (Figure 7).

After examining the impact of IGFBP-2 on DNA damage induced by etoposide in estrogen-receptor-positive breast cancer cells, the study extended its investigation to triple-negative breast cancer cells (TNBC), characterised by the absence of ER, PR, and HER2 (a member of the EGFR family) [40,43]. Notably, when exposed to exogenous recombinant IGFBP-2, the MDA-MB-231 cells exhibited enhanced DNA repair mechanisms and an upregulation of DNA-PKcs leading to greater resistance to etoposide, and suggested a protective role for IGFBP-2. This finding indicates that IGFBP-2 can interact with DNA-PKcs in the absence of ER and aligns with the hypothesis that IGFBP-2 mitigates DNA damage by enhancing the DNA repair mechanism. To determine whether exogenous IGFBP-2 was acting intrinsically through integrin receptors, exogenous IGFBP-2 was added with or without a disintegrin peptide, RGD. Strikingly, the study revealed that RGD had a similar effect to exogenously added IGFBP-2, resulting in improved DNA repair efficiency and reduced DNA damage in treated cells compared to those treated with etoposide alone. This suggests that IGFBP-2 and RGD may have a similar mechanism of action, likely dependent on integrin signalling, but whether IGF influences this is yet to be determined. Integrin signalling has been shown to promote DNA repair via NHEJ [44], and it is widely reported that the IGF-independent actions of IGFBP-2 are via activation of integrin signalling [45,46] (Figure 7).

TNBC cells often overexpress the IGF-1 and EGF receptors [47] and it has been demonstrated that they can translocate to the nucleus, where they have been implicated in modulating the DNA damage response [13,48,49,50,51,52]. In addition, IGFBP-2 demonstrates associations not only with the IGF-1R through binding IGFs but also notably with EGFR. For example, IGFBP-2 was found in complex with EGFR, assisting EGFR accumulation in the nucleus [53,54,55,56,57].

In ER-positive cells, silencing IGFBP-2 reduced DNA-PKcs and this enhanced sensitivity to etoposide. We showed previously that silencing IGFBP-2 is associated with loss of the ERα [19], which is a key regulator of DNA-PKcs [39]. The relationship between IGFBP-2 and ERα may be one explanation as to why loss of IGFBP-2 in MCF-7/T47D cells results in reduced levels of DNA-PKcs leading to an enhanced response to etoposide. In ER-negative cells, IGFBP-2 interacted with integrin receptors that led to its internalisation and translocation to the nucleus, which resulted in an increase in levels of DNA-PKcs and chemoresistance. Receptor tyrosine kinase receptors, including the EGF and IGF receptors, commonly translocate to the nucleus [58]. Previous studies reported that in esophageal cancer cells, IGFBP-2 forms a nuclear complex with EGFR and DNA-PKcs following DNA damage [20]. The exact mechanism by which IGFBP-2 modulates DNA-PKcs requires further investigation, but it seems likely that this will be context-dependent.

Abbreviations: Estradiol, E2; Estrogen receptor-α, ERα; Mitogen-activated protein kinases, MAPK; Ak strain transforming AKT; phosphatidylinositol 3-kinase, PI3K, Phosphatase and tensin homologue, PTEN (Produced in Biorender).

## 5. Conclusions

This work contributes to our understanding of the role of IGFBP-2 in chemoresistance and suggests that it plays a part in promoting DNA repair through modulation of DNA-PKC. The mechanism by which IGFBP-2 achieves this and the involvement of IGF-I/IGF-1R require further investigation and are likely to be context-dependent.

## Figures and Tables

**Figure 1 cancers-16-02113-f001:**
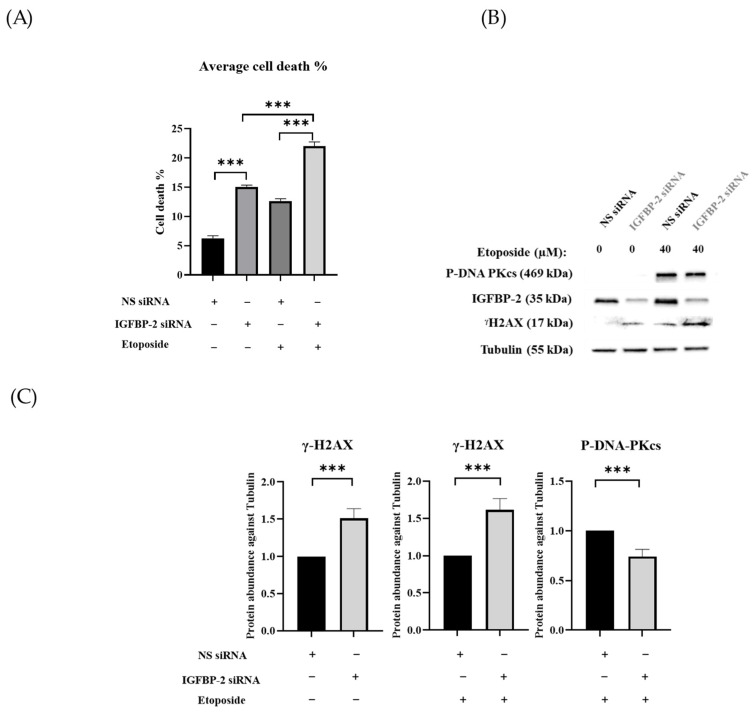
Shows MCF-7 and T47D cells treated with or without etoposide in the presence or absence of IGFBP-2. (**A**,**D**) show the mean of percentage cell death and (**B,E**) Western blot analysis of cell lysates of MCF-7 and T47D cells, respectively, with and without IGFBP-2, in the presence or absence of etoposide (40 µM and 60 µM, respectively) for 24 h. (**C**,**F**) Optical densities of lysates of treated MCF-7 and T47D cells, respectively, after correcting to tubulin and normalising to 1; tubulin was used as a loading control (where * *p* < 0.05, ** *p* < 0.01 and *** *p* < 0.001). Cell counting was performed three times, each repeated in triplicate, and graphs show the mean +/− SEM. The Western blots are representative of experiments repeated three times, from which the mean fold change +/− SEM optical density measurements are shown. Original western blots are presented in Appendix A.

**Figure 2 cancers-16-02113-f002:**
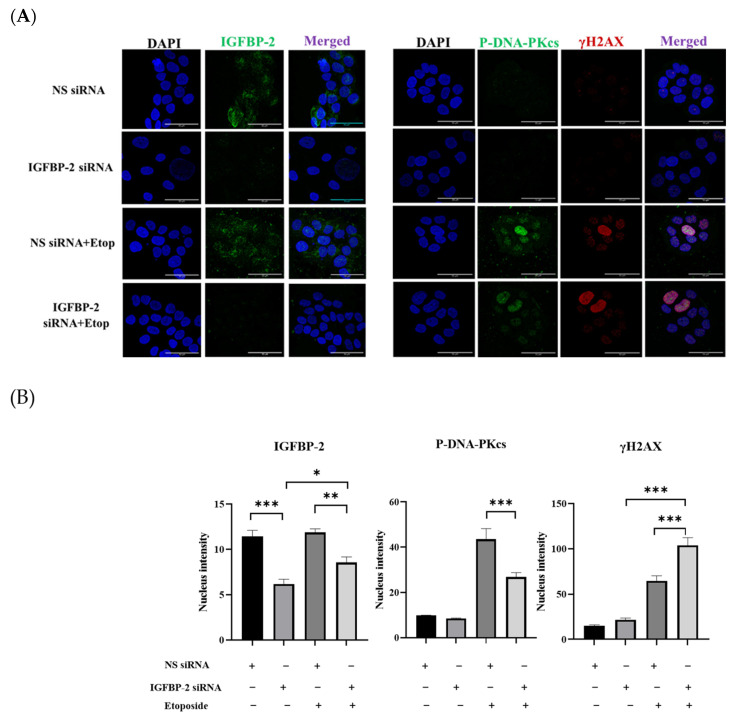
(**A**) Immunofluorescence staining of IGFBP-2 (green), P-DNA-PKcs (green), and γH2AX (red) of T47-D cells (scale bar = 50 μm). Three independent experiments were performed with each repeated in triplicate. (**B**) Quantification of intensity of immunofluorescence in T47D cells within the nucleus for IGFBP-2, P-DNA-PKcs, and γH2AX. Immunofluorescence images were analysed by Fiji-image J program, which measured the fluorescence intensity of each protein inside the nucleus. Three independent experiments were performed with each repeated in triplicate, from which the mean of the nuclear density +/− SEM measurements are shown (where * *p* < 0.05, ** *p* < 0.01 and *** *p* < 0.001).

**Figure 3 cancers-16-02113-f003:**
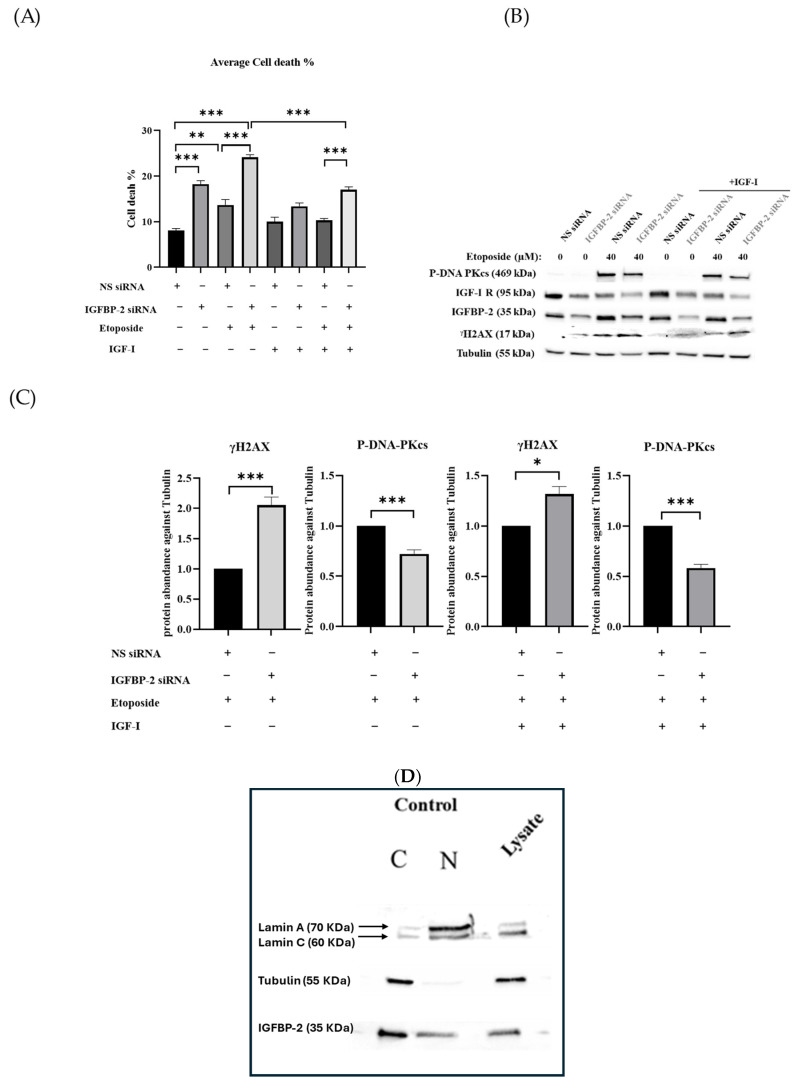
IGFBP-2 silencing in MCF-7 cells with or without IGF-I and etoposide. (**A**) Shows the mean of percentage cell death. (**B**) Western blot analysis of cell lysates of MCF-7 cells when IGFBP-2 siRNA is present and absent and 10 ng/mL IGF-I treated with and without etoposide (40 µM) for 23 h. As a loading control, tubulin was used. (**C**) Optical densities of MCF-7 lysates of treated cells expressed as relative fold change after correcting to tubulin and normalising to 1 (where * *p* < 0.05, ** *p* < 0.01 and *** *p* < 0.001) of three independent experiments each repeated in triplicate. (NS = non-silencing). One-way ANOVA was used for statistical analysis. (**D**) Nuclear (N) and cytoplasmic (C) fractions of MCF-7 cells were analysed by Western blotting. As nuclear and cytoplasmic loading controls, lamin A/C and tubulin were used, respectively. (**E**–**G**) Immunofluorescence staining of IGFBP-2 (green), P-DNA-PKcs (green), and γH2AX (red) in MCF-7 cells. FIJI/Image J was used to process the images (scale bar = 50 μm). Three independent experiments were performed with each repeated in triplicate. (**H**) Quantification of intensity of immunofluorescence within the nucleus for IGFBP-2, P-DNA-PKcs, and γH2AX in MCF-7 cells. Immunofluorescence images was analysed by Fiji-image J program, which measures the fluorescence intensity of each protein inside the nucleus. Three independent experiments were performed with each repeated in triplicate. Original western blots are presented in Appendix A.

**Figure 4 cancers-16-02113-f004:**
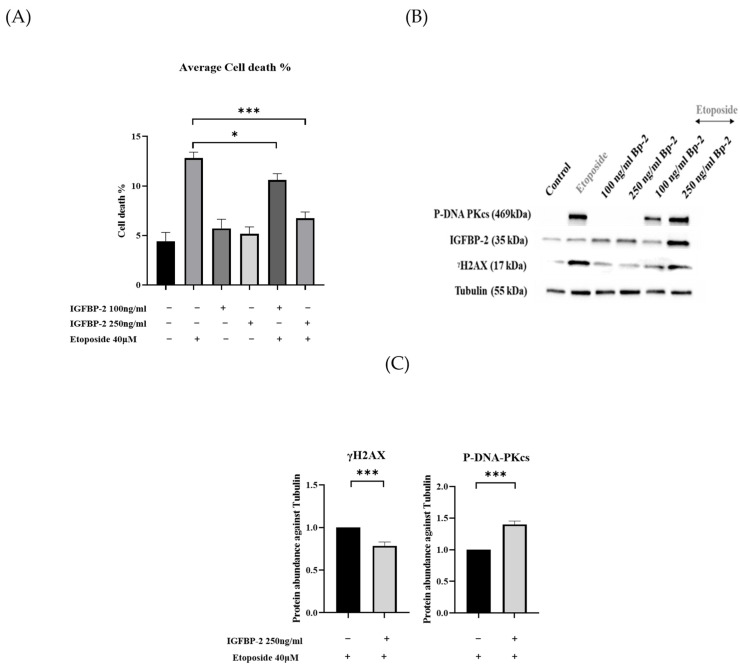
MDA-MB-231 cells dosed with recombinant IGFBP-2 with or without etoposide 40 µM. (**A**) Shows mean of percentage cell death in MDA-MB-231 cells dosed with recombinant IGFBP-2 (100 and 250 ng/mL) with and without etoposide 40 µM. (**B**) Western blot analysis of cells lysates. (**C**) Optical density analysis of γH2AX and P-DNA-PKcs in MDA-MB-231-treated cell lysates. Protein abundance was corrected for tubulin and normalised to cells treated with etoposide alone (where * *p* < 0.05, and *** *p* < 0.001). The results shown are the mean +/− SEM of three independent experiments each repeated in triplicate. Original western blots are presented in Appendix A.

**Figure 5 cancers-16-02113-f005:**
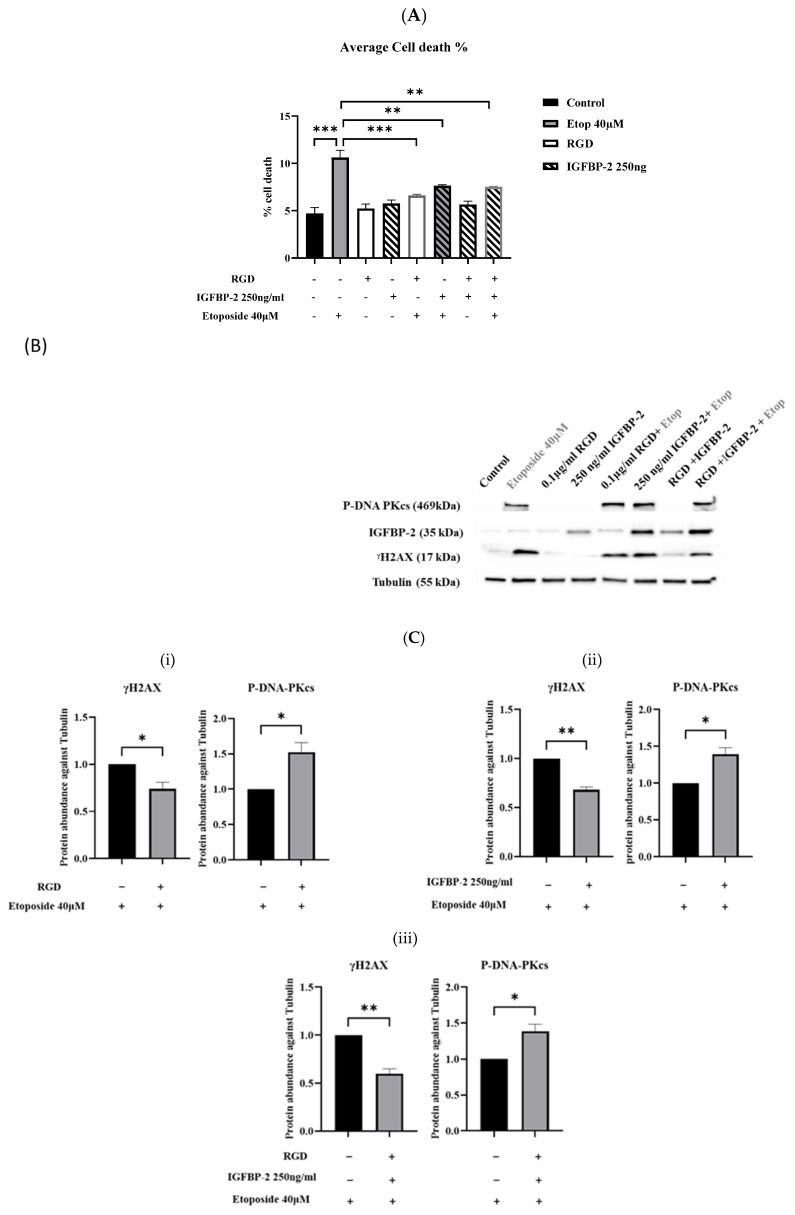
MDA-MB-231 cells dosed with recombinant IGFBP-2 with or without etoposide in the presence or absence of RGD. (**A**) Shows mean of percentage cell death in MDA-MB-231 cells pre-dosed with recombinant IGFBP-2 (250 ng/mL) for 4 h and RGD (0.1 µg/mL) for 1 h, treated with and without etoposide (40 µM) for 19 h. Results shown are the mean +/− SEM of three independent experiments each repeated in triplicate. (**B**) Representative Western blot analysis of cell lysates repeated three times. (**C**) Optical densities to measure γH2AX and P-DNA-PKcs in MDA-MB-231 cell lysate treated with (**i**) RGD and etoposide, (**ii**) IGFBP-2 and etoposide, and (**iii**) IGFBP-2 and RGD compared with etoposide alone, after correcting for tubulin and normalising to 1 (where * *p* < 0.05, ** *p* < 0.01 and *** *p* < 0.001). Original western blots are presented in Appendix A.

**Figure 6 cancers-16-02113-f006:**
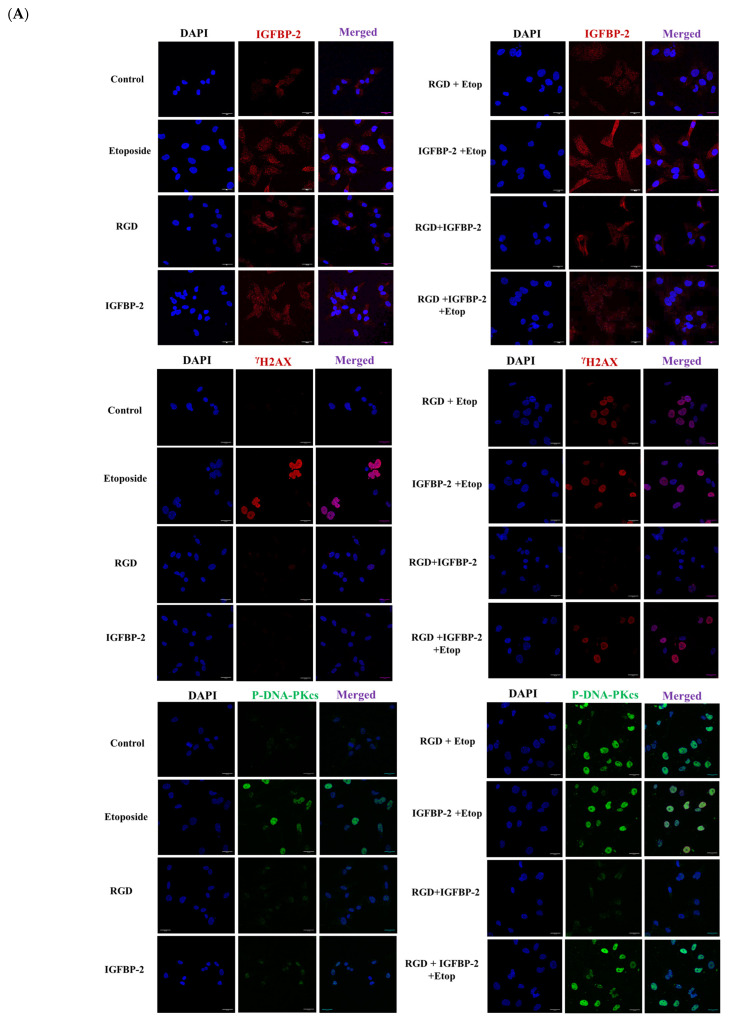
Immunofluorescence staining of MDA-MB-231 cells in the presence and absence of recombinant IGFBP-2 (250 ng/mL) for 4 h and RGD (0.1 µg/mL) for 1 h, treated with and without etoposide (40 µM) for 19 h (scale bar = 50 μm). (**A**) Immunofluorescence staining of IGFBP-2 (red), γH2AX (red), and P-DNA-PKcs (green). (**B**) Quantification intensity of immunofluorescence within the nucleus for IGFBP-2, γH2AX, and P-DNA-PKcs. Results shown are the mean of three independent experiments. Immunofluorescence images were analysed by Fiji-image J program that measures the fluorescence intensity of each protein inside the nucleus (where *** *p* < 0.001).

**Figure 7 cancers-16-02113-f007:**
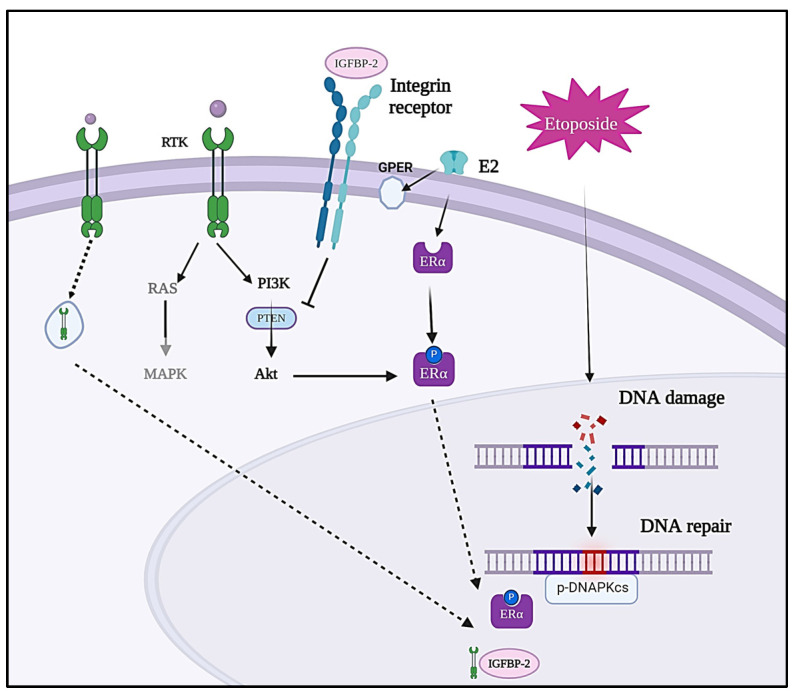
Proposed involvement of IGFBP-2 in etoposide-induced DNA damage in breast cancer cells.

## Data Availability

No new data were collected.

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
