# Peer review of "The Role of Insulin-like Growth Factor Binding Protein (IGFBP)-2 in DNA Repair and Chemoresistance in Breast Cancer Cells"

_cancers, 2024, doi:10.3390/cancers16112113_

Round 1
Reviewer 1 Report
Comments and Suggestions for Authors
Mohammedali. et al. investigated the role of insulin-like growth factor binding protein-2 (IGFBP-2) in influencing the response of different breast cancer cell lines to etoposide. The authors claimed that in ER positive cells, silencing IGFBP-2 increases cell death and affects DNA repair mechanisms, whereas in ER negative cells, adding IGFBP-2 promotes cell survival, potentially by enhancing DNA repair processes. While the results are somewhat interesting, the authors may need more direct evidence to support the conclusion.
1. Throughout the manuscript, trypan blue exclusion is the only experimental readout for cell death. This is a very crude estimation and more careful assessment of cell death should be done, like Annexin V and PI staining by flow cytometry.
2. Similarly, to claim a phenotype in survival, trypan blue exclusion is far from satisfactory. Colony forming assay should be performed under different conditions.
3. The IF image of γH2AX is not ideal, probably a result of overexposure, as real γH2AX damage signals should be foci not a whole light up nuclei.
4. The authors should consider adding a model figure, either cited at the end of the result section or in the discussion when explaining the potential mechanism.
5. Is the effect of IGFBP-2 on DNA repair specific to S phase damage induced by etoposide? What about IR?
Author Response
We thank the reviewer for their constructive comments and suggestions and have highlighted changes to the manuscript in green.
- Throughout the manuscript, trypan blue exclusion is the only experimental readout for cell death. This is a very crude estimation and more careful assessment of cell death should be done, like Annexin V and PI staining by flow cytometry.
- Similarly, to claim a phenotype in survival, trypan blue exclusion is far from satisfactory. Colony forming assay should be performed under different conditions.
We thank the reviewer for these methodological suggestions. We previously rigorously characterized chemotherapy-induced apoptosis and survival in these cell lines using flow cytometry, MTT assay, PARP cleavage, colony formation assay and morphological assessment and confirmed that all measures compared well with TB as a read out of cell death. We have added a sentence to the methods (lines 98-101) to highlight this point with supportive references.
- The IF image of γH2AX is not ideal, probably a result of overexposure, as real γH2AX damage signals should be foci not a whole light up nuclei.
We acknowledge this point and agree that in some of the images, whilst foci are evidently seen in some nuclei, in other nuclei they are less distinguishable. We are confident that the western blot analysis using the γH2AX antibody does mirror the changes we see when performing IF analysis, and these correlate with levels of cell death.
- The authors should consider adding a model figure, either cited at the end of the result section or in the discussion when explaining the potential mechanism.
Thank you for this suggestion, we have now added a schematic to suggest how IGFBP-2 may contribute to DNA repair following chemotherapy(Figure 7)(Inserted at line 432)
- Is the effect of IGFBP-2 on DNA repair specific to S phase damage induced by etoposide?
Thank you for this question. As etoposide blocks the cell cycle in the late S or early G2 phase, we suggest that this is when IGFBP-2 may act. However, we could not specify this in the paper as levels of cyclins for example, were not examined.
- What about IR? We did not examine the effects of ionizing radiation on the cells, but this would be an interesting comparison.
Reviewer 2 Report
Comments and Suggestions for Authors
The study is focused on understanding the potential role of IGFBP-2 in DDR in breast cancer cells.
Figure 1
1. Use of trypan blue exclusion for cell viability although acceptable - use of more than one method of cell viability is more accurate (clonogenic,MTT,CTG)
2. Please include molecular weights in WB
3. Probe for other markers of DDR - p-RPA; p-KAP1 ( along with total RPA and KAP1) etc.
4. Use a scale of 100% for all cell death data - retains consistency across the paper.
H2AX IF
1. H2AX focii formation is a more important marker to study from a DDR perspective.
- Validation using multiple siRNAs is important to solidify the claims.
- Dose-response of etopside that lead to the identification of this dose is mssing ...
- Other assays to study DDR like micronuclei formation are important to consider.
- How does BRCA status alter the hypothesis presented here?
- How does the loss of any IGFBP-binding proteins alter the results? Maybe an important perspective to consider for studying the pathway and any future biomarkers.
Author Response
We thank the reviewer for their constructive comments and suggestions and have highlighted changes to the manuscript in yellow.
The study is focused on understanding the potential role of IGFBP-2 in DDR in breast cancer cells.
- Use of trypan blue exclusion for cell viability although acceptable - use of more than one method of cell viability is more accurate (clonogenic,MTT,CTG).
We thank the reviewer for these methodological suggestions. We previously rigorously characterized chemotherapy-induced apoptosis and survival in these cell lines using flow cytometry, MTT assay, PARP cleavage, colony formation assay and morphological assessment and confirmed that all measures compared well with TB as a read out of dead cells. We have added a sentence to the methods (lines 98-101) to highlight this point with supportive references.
- Please include molecular weights in WB
The molecular weights of each of the proteins is included in each of the blots- the original blots and ladders were also submitted to the journal for verification of protein sizes shown in the paper.
- Probe for other markers of DDR - p-RPA; p-KAP1 ( along with total RPA and KAP1) etc.
Thank you for the suggestions of measuring additional markers of DDR but this was beyond the scope of this PhD study.
- Use a scale of 100% for all cell death data - retains consistency across the paper.
Thank you for the suggestion of keeping the scale for cell death the same on all graphs- we had attempted to do this and feel we should keep it the same, if the reviewer is in agreement, as changing to 100% would make the differences difficult to see.
- H2AX focii formation is a more important marker to study from a DDR perspective.
Thank you for this clarification.
- Validation using multiple siRNAs is important to solidify the claims.
We did validate the IGFBP-2 siRNA using two different siRNAs that are referenced in the methods (Line 95).
- Dose-response of etoposide that lead to the identification of this dose is missing.
We performed rigorous testing to determine the optimum dose for etoposide in all the cell lines. This included an assessment of different doses (40, 60 & 80 µM) and time courses (1, 4, 24 & 48 hrs) with the measurement of γH2AX abundance as a marker of DNA damage. We have now described this in the methods (lines:86-90)
Treatment with Etoposide: MCF-7 and MDA-MB-231 cells were treated with (40 µM) and T47D (60 µM) etoposide for 24 hrs. We performed rigorous testing to determine the optimum dose for etoposide in all the cell lines. This included an assessment of different doses (40, 60 & 80 µM) and time courses (1, 4, 24 & 48hrs) with the measurement of γH2AX abundance as a marker of DNA damage (data not shown).
- Other assays to study DDR like micronuclei formation are important to consider.
Thank you for this suggestion, that we shall certainly consider in future studies.
- How does BRCA status alter the hypothesis presented here?
The cell lines used in this study are BRCA wt. This is an interesting question, but not one we specifically addressed in this study.
Interestingly, there have been numerous reports indicating that BRCA1, a key tumour suppressor gene involved in DNA repair, can bind to the ER and inhibit estrogen/ER signaling (e.g. doi: 10.7150/ijbs.8579.) (https://doi.org/10.1038/sj.onc.1208190). Estrogen/ER signalling is known to upregulate the expression of DNAPKCs (https://doi.org/10.1038/embor.2009.279) and we showed previously that loss of IGFBP-2 results in loss of ERα (doi: 10.1210/en.2012-1970.). Although not investigated, this may also affect the stability of BRCA, suggesting one way in which IGFBP-2 and BRCA1 may be linked in ER-positive cells to modulate DNAPKcs and the response to DNA damage.
9 How does the loss of any IGFBP-binding proteins alter the results? Maybe an important perspective to consider for studying the pathway and any future biomarkers.
This is a good question but was not addressed in this study. We did show previously that basally in MCF-7 cells the predominant IGFBP is IGFBP-2 [https://doi.org/10.1038/sj.onc.1210397]. However, compensation with modulation of other IGFBPs, upon treatment may be a possibility. As the reviewer suggests this will be an important consideration going forwards when more information is available on the potential role of other IGFBPs (that have nuclear localization sequences) in the DNA damage response and how they may work in concert. Evidence has shown for example that IGFBP-3 plays a role in the DNA damage response, through an EGFR-IGFBP-3 interaction in triple negative breast cancer cell lines, MDA-MB-468 and Hs578T (https://doi.org/10.1038/onc.2012.538).
Reviewer 3 Report
Comments and Suggestions for Authors
In this manuscript Mohammedali et al reported a likely association between IGFBP-2 and DNA damage response in breast cancer cell lines. The manuscript appears to be a preliminary report which consists of repeats of the same set of experiments, and lacks insightful discussion and conclusions.
1. The role of IGFBP-2 in DNA damage response is not as novel as mentioned in current manuscript since it’s already reported (PMID: 30626422), where IGFBP-2 is proposed to activate EGFR-DNA-PKcs pathway.
2. The current manuscript is limited to cell death analysis and protein expression. It could be solidified, for example, by in vivo experiments showing if targeting IGFBP-2 can improve chemotherapy response. Cell proliferation assay could also be included besides cell death measurement.
3. As IGFBP-2 is a secreted protein, have authors examined protein level in culture medium? Authors added exogeneous IGFBP-2 into the medium of MDA-MB-231. How this treatment changes medium composition should be analyzed. Besides, authors suggest that IGFBP-2 may act in a way similar to integrin pathway. This also suggests extracellular IGFBP-2 level may be more relevant.
4. Authors only did exogenous addition of IGFBP-2 in MDA-MB-231, an ER negative cell line, as it has a lower endogenous IGFBP-2 level. However, it can be informative to include exogeneous IGFBP-2 in the other two cell lines, or to conduct knock-down experiments in MDA-MB-231 cells which still express detectable IGFBP-2 protein (Fig. 4B).
Author Response
We thank the reviewer for their constructive comments and suggestions and have highlighted changes to the manuscript in yellow.
In this manuscript Mohammedali et al reported a likely association between IGFBP-2 and DNA damage response in breast cancer cell lines. The manuscript appears to be a preliminary report which consists of repeats of the same set of experiments and lacks insightful discussion and conclusions.
- The role of IGFBP-2 in DNA damage response is not as novel as mentioned in current manuscript since it’s already reported (PMID: 30626422), where IGFBP-2 is proposed to activate EGFR-DNA-PKcs pathway.
Thank you for this observation and we agree with the reviewer that IGFBP-2 has been linked with DNA damage previously in esophageal adenocarcinoma cells from acidic bile salts-induced DNA damage. The novel aspect related to such a role for IGFBP-2 in etoposide-induced DNA damage in breast cancer cell. However, to acknowledge this comment we have removed ‘novel’ from the title.
The current manuscript is limited to cell death analysis and protein expression. It could be solidified, for example, by in vivo experiments showing if targeting IGFBP-2 can improve chemotherapy response. Cell proliferation assay could also be included besides cell death measurement.
In vivo work was beyond the funding scope of this PhD study. We have included some data on cell proliferation as a supplementary figure 1 and added the information to the text on line 203.
‘Further, cell growth was significantly inhibited in the presence of IGFBP-2 siRNA (p<0.01) alone but not by etoposide alone and there was no additive effect in combination. This pattern was unaffected in the presence of IGF-I (Supplementary figure 1A).’
And on line 312.
‘There were no significant reductions in total cell number with any treatment (Supplementary figure 1B).’
- As IGFBP-2 is a secreted protein, have authors examined protein level in culture medium? Authors added exogeneous IGFBP-2 into the medium of MDA-MB-231. How this treatment changes medium composition should be analyzed. Besides, authors suggest that IGFBP-2 may act in a way similar to integrin pathway. This also suggests extracellular IGFBP-2 level may be more relevant.
In addition to measuring IGFBP-2 in the cell lysate, we also measured it in the supernatant of the MDA-MB-231 cells. The lysates and cell supernatants clearly indicate the exogenous addition of IGFBP-2. We have added an example to supplementary figure 1- panel C and referred to this in the text on line 286.
- Authors only did exogenous addition of IGFBP-2 in MDA-MB-231, an ER negative cell line, as it has a lower endogenous IGFBP-2 level. However, it can be informative to include exogeneous IGFBP-2 in the other two cell lines, or to conduct knock-down experiments in MDA-MB-231 cells which still express detectable IGFBP-2 protein (Fig. 4B).
Thank you for this suggestion, we agree that the converse experiments may be useful to confirm the data, but this was beyond the time frame of the study, and we have now provided rationale for this approach on lines 143 and 279.
Reviewer 4 Report
Comments and Suggestions for Authors
A novel role for insulin-like growth factor binding protein 2 (IGFBP)-2 in DNA repair in breast cancer cells
A brief summary:
This study aims to investigate how IGFBP-2 influences the DNA damage and repair in breast cancer cells as well as explore its interaction with IGFs through dependent or independent actions. The results shows that the loss of IGFBP-2 alone induced DNA damage due to the increase levels of γH2AX in the presence of etoposide. Silencing IGFBP-2 causes the reduced levels of P-DNA-PKcs, which is related to decreased function of DNA repair.
General concept comments:
The conclusions are supported by the results but some improvements are needed for the result part.
· Line 132, the statement “and silencing IGFBP-2 with etoposide had an additive effect (Figure 1B&C)”. Based on Figure 1B, it did show the additive effect in silencing IGFBP-2 with 40 µM Etoposide as the γH2AX abundance is the highest in Western blot analysis. However, when comes to Figure 1C comparing the protein abundance against Tubulin on γH2AX, the result in silencing IGFBP-2 with Etoposide is close to the result with silencing IGFBP-2 only(not showing the additive effect), would you explain more or provide a clearer guidance between your statement with the figures?
· Line 176, scale bar unit should be 50 µm not µM.
· Line 191-192, the statement “IF confirmed that IGFBP-2 was effectively reduced in the presence of IGFBP-2 siRNA (Figure 3D)” is confused here. IF should be Immunofluorescence? It doesn’t include the abbreviation in the entire paper. In addition, please provide the correct figures to support this statement. Figure 3D is the western blot which shows IGFBP-2 was observed in both the cytoplasm and nuclear extracts. Immunofluorescence figures are Figure 3E-G.
· Line 193-194, please provide the correct figures to support this statement. The result of Figure 3H is missing in this paragraph.
· Line 238-240, the description in figure content of Figure 3(D), “(D) Nuclear (N) and cytoplasmic (C) fractions of MCF-7 cells were analyzed by western blotting in the presence and absence of etoposide (40 µM). As nuclear and cytoplasmic loading controls, lamin A/C and tubulin were used respectively” didn’t match/or difficult to follow in Figure 3(D). Please clarify which is the presence and absence of etoposide in the western blot. The original blot of Figure 3D needs additional information to clarify the bands. In addition, Line # 203-207 were collapsed together with Figure 3(D) which needs to be adjusted.
· Line 284, in Figure 5A, in the presence of etoposide, the cell death % also shows a difference between RGD and exogenous IGFBP-2 (bar 5 and bar 6 in figure 5A) based on the error bar is not overlapping between these two. Would you explain what might be the possibility for the difference?
· Line 289, it will be better to split Figure 5B &C in each corresponding statement instead of putting them together after two statements, otherwise, it will be difficult to follow those figures. For example: line 286-288, RGD and IGFBP-2 each in combination with etoposide resulted in a reduction in γH2AX and an increase in abundance of P-DNA-PKcs (Figure 5C). These effects were not enhanced when RGD and IGFBP-2 were added in combination with etoposide (which figure is supporting this statement?). “These effects were not enhanced”, do you mean the additive effect? Please specify. Would you provide some possibilities why these effects are not enhanced in the discussion section? In addition, because there are so many figures in the entire results sections, it would be better to provide a clearer guidance between figure and statement/fact.
· Line 329-330. Figure 6B, when comparing the change in level of P-DNA-PKcs, why adding RGC, IGFBP-2 together with Etoposide doesn’t not have the additive effect in the nucleus intensity?
· Line 372. “Additionally, the research indicated that estrogen induces the expression of DNA-PKcs in breast cancer cells potentially facilitating DNA damage repair.” Please specify which reference supports this.
Author Response
We thank the reviewer for their constructive comments and suggestions and have highlighted changes to the manuscript in yellow.
The conclusions are supported by the results, but some improvements are needed for the result part.
- Line 132, the statement “and silencing IGFBP-2 with etoposide had an additive effect (Figure 1B&C)”. Based on Figure 1B, it did show the additive effect in silencing IGFBP-2 with 40 µM Etoposide as the γH2AX abundance is the highest in Western blot analysis. However, when comes to Figure 1C comparing the protein abundance against Tubulin on γH2AX, the result in silencing IGFBP-2 with Etoposide is close to the result with silencing IGFBP-2 only (not showing the additive effect), would you explain more or provide a clearer guidance between your statement with the figures?
Thank you. As you mentioned, the blot clearly shows the additive effect in silencing IGFBP-2 with 40 µM Etoposide. The OD graph (far left) shows the effect of IGFBP-2 siRNA alone on γH2AX with the data corrected to NS γH2AX, showing the effect of silencing IGFBP-2. The blot indicates that etoposide alone has an effect on its own and the middle graph has corrected the effect of etoposide alone to 1, to show that IGFBP-2 can further increase the impact of etoposide.
- Line 176, scale bar unit should be 50 µm not µM.
Thank you for spotting this error-the figure legends have been amended accordingly.
- Line 191-192, the statement “IF confirmed that IGFBP-2 was effectively reduced in the presence of IGFBP-2 siRNA (Figure 3D)” is confused here. IF should be Immunofluorescence? It doesn’t include the abbreviation in the entire paper.
We have now referred to immunofluorescence in full throughout the manuscript.
In addition, please provide the correct figures to support this statement. Figure 3D is the western blot which shows IGFBP-2 was observed in both the cytoplasm and nuclear extracts. Immunofluorescence figures are Figure 3E-G. · Line 193-194, please provide the correct figures to support this statement. The result of Figure 3H is missing in this paragraph.
We changed D to E on line 209, to refer to the correct image and, also added clarity to lines 210-212 to indicate the correct figures (immunofluorescence and associated OD analysis). Thank you for noticing this oversight.
- Line 238-240, the description in figure content of Figure 3(D), “(D) Nuclear (N) and cytoplasmic (C) fractions of MCF-7 cells were analyzed by western blotting in the presence and absence of etoposide (40 µM). As nuclear and cytoplasmic loading controls, lamin A/C and tubulin were used respectively” didn’t match/or difficult to follow in Figure 3(D). Please clarify which is the presence and absence of etoposide in the western blot. The original blot of Figure 3D needs additional information to clarify the bands. In addition, Line # 203-207 were collapsed together with Figure 3(D) which needs to be adjusted.
Thank you so much for noticing this oversight. The figure does not show any effects of etoposide and just shows basal localization of IGFBP-2. We have amended the figure legend accordingly (line 269). In addition, we have re-labelled this panel for clarity.
- Line 284, in Figure 5A, in the presence of etoposide, the cell death % also shows a difference between RGD and exogenous IGFBP-2 (bar 5 and bar 6 in figure 5A) based on the error bar is not overlapping between these two. Would you explain what might be the possibility for the difference.
An interesting point, thank you. Yes, there does appear to be different levels of inhibition with RGD + etoposide compared with exogenously added IGFBP-2+ etoposide. This has not been investigated but comparing effects of relative levels of a small peptide with the same peptide within IGFBP-2 is difficult in terms of accuracy and also in terms of other molecules with which IGFBP-2 could potentially bind.
- Line 289, it will be better to split Figure 5B &C in each corresponding statement instead of putting them together after two statements, otherwise, it will be difficult to follow those figures. For example: line 286-288292, RGD and IGFBP-2 each in combination with etoposide resulted in a reduction in γH2AX and an increase in abundance of P-DNA-PKcs (Figure 5C). These effects were not enhanced when RGD and IGFBP-2 were added in combination with etoposide (which figure is supporting this statement?). Figure 5B & C). “These effects were not enhanced”, do you mean the additive effect? Please specify.
Thank you for this suggestion. We have re-labelled the figure and split panel C into three parts (i, ii & iii) and used this notation to make the text in the results clearer (line 321 & 322). We have also indicated that we are referring to the lack of an additive effect ‘There was no additive effect when RGD and IGFBP-2 were added in combination with etoposide (Figure 5B & Ciii)’ (line 321) . We have amended the figure legend accordingly (line 345 & 346).
Would you provide some possibilities why these effects are not enhanced in the discussion section? In addition, because there are so many figures in the entire results sections, it would be better to provide a clearer guidance between figure and statement/fact.
We have added some additional information to the discussion on lines 427-431. To provide further clarity we have included a schematic to suggest the role for IGFBP-2 in DNA repair (Figure 7).
Line 329-330. Figure 6B, when comparing the change in level of P-DNA-PKcs, why adding RGC, IGFBP-2 together with Etoposide doesn’t not have the additive effect in the nucleus intensity?
This result is consistent with the western blotting data showing no extra impact of adding RGD and IGFBP-2 with etoposide compared with RGD+ etoposide or IGFBP-2+etoposide.
Line 372. “Additionally, the research indicated that estrogen induces the expression of DNA-PKcs in breast cancer cells potentially facilitating DNA damage repair.” Please specify which reference supports this.
Thank you for questioning these references, we have now added a specific reference to support this statement for clarity :ref 44.
Reviewer 5 Report
Comments and Suggestions for Authors
The manuscript titled “A novel role for insulin-like growth factor binding protein (IGFBP)-2 in DNA repair in breast cancer cells” is well designed and conducted. The results of this work indicated the significance of IGFBP-2 in the mechanism of resistance to cancer therapy in some types of cancers. I recommend publishing this manuscript after performing the following minor suggestions:
The title is very broad so it needs to be very specific in indicating the role of IGFBP-2 in cancer resistance
IGFBP-2 was silenced in ER+ve cells and added to ER-ve cells. Why not silenced in both cells and/or added both cells. The rationale should be clear here.
In line 65-69. The authors should expand on the previously found role of IGFBP-2 in literature. Some results related to this work is not mentioned and referenced. For example in Endocrinology. 2013 May, IGFBP-2 was previously silenced in MCF-7. Therefore, the literature review should be thorough in this point and linked to the rationale of this study.
In the methodology, the cytotoxicity and western blotting need to be written in detail.
Author Response
We thank the reviewer for their constructive comments and suggestions and have highlighted changes to the manuscript in yellow.
The manuscript titled “A novel role for insulin-like growth factor binding protein (IGFBP)-2 in DNA repair in breast cancer cells” is well designed and conducted. The results of this work indicated the significance of IGFBP-2 in the mechanism of resistance to cancer therapy in some types of cancers. I recommend publishing this manuscript after performing the following minor suggestions:
The title is very broad, so it needs to be very specific in indicating the role of IGFBP-2 in cancer resistance.
Thank you for this suggestion, we have amended the title accordingly ’ A role for insulin-like growth factor binding protein (IGFBP)-2 in DNA repair and chemoresistance in breast cancer cells’.
IGFBP-2 was silenced in ER+ve cells and added to ER-ve cells. Why not silenced in both cells and/or added both cells. The rationale should be clear here.
Thank you for this suggestion, we have now provided the rationale for our approach to manipulating IGFBP-2 in the different cell lines (143-144 & 279-281).
In line 65-69. The authors should expand on the previously found role of IGFBP-2 in literature. Some results related to this work is not mentioned and referenced. For example in Emily J Foulstone et al. Endocrinology. 2013 May, IGFBP-2 was previously silenced in MCF-7. Therefore, the literature review should be thorough in this point and linked to the rationale of this study.
Thank you for this suggestion this has been cited correctly now in the introduction on lines 68-70.
In the methodology, the cytotoxicity and western blotting need to be written in detail.
We have added additional detail to the methods as requested covering cytotoxicity and western blotting (Lines 86-90, 98-101 & 105-113)
Round 2
Reviewer 1 Report
Comments and Suggestions for Authors
No further comments
Reviewer 2 Report
Comments and Suggestions for Authors
The authors have addressed most of my concerns.
Reviewer 3 Report
Comments and Suggestions for Authors
All my points are addressed. Please align sub panels in each figure to improve readability.